# Reduced Meal Frequency Decreases Fat Deposition and Improves Feed Efficiency of Growing–Finishing Pigs

**DOI:** 10.3390/ani12192557

**Published:** 2022-09-24

**Authors:** Shanchuan Cao, Wenjie Tang, Hui Diao, Shuwei Li, Honglin Yan, Jingbo Liu

**Affiliations:** 1School of Life Science and Engineering, Southwest University of Science and Technology, Mianyang 621010, China; 2Department of Animal Resource and Science, Dankook University, Cheonan 31116, Korea; 3Livestock and Poultry Biological Products Key Laboratory of Sichuan Province, Sichuan Animtech Feed Co., Ltd., Chengdu 610066, China

**Keywords:** meal frequency, growth performance, lipid metabolism, growing pig

## Abstract

**Simple Summary:**

In this research, we studied the effects of feeding frequency (three meals per day vs. free access to feed) on growth performance, nutrient digestibility, carcass quality, and lipid metabolism in growing–finishing pigs. The results showed that reducing feeding frequency can effectively improve the feed conversion ratio. The three meals per day group had a different growth performance in terms of reduced fat deposition and changed carcass composition. This study provides a new scheme for the feeding and management of growing–finishing pigs.

**Abstract:**

An experiment was conducted to examine the effect of meal frequency on growth performance, nutrient digestibility, carcass quality, and lipid metabolism in growing–finishing pigs. Sixty-four Duroc × Landrace × Yorkshire barrows and gilts (26.40 ± 2.10 kg initial body weight) were used in a 112-d experiment in a randomized complete blocked design. The two treatments were the free-access feed group (FA) and the three meals per day group (M3), respectively. The result showed that the average daily feed intake (ADFI) and F: G of the FA group were significantly higher than that in the M3 group during the whole experiment (*p* < 0.05). Reducing meal frequency also decreased the concentration of triglycerides and urea nitrogen but increased the concentration of insulin and free fatty acids in the blood (*p* < 0.05). Reducing meal frequency decreased compositions of backfat, belly, and fatty pieces but increased compositions of ham, longissimus muscle, and lean pieces in the carcass (*p* < 0.05). Greater enzyme activities of ME and FAS and higher mRNA expression of *FAS* and *PPARγ* were found in the LM of FA pigs compared with M3 pigs (*p* < 0.05). In summary, a lower meal frequency improves feed efficiency by regulating lipid metabolism and reducing fat deposition.

## 1. Introduction

The total cost of pig production consists of the total feed cost, total fixed cost, total reproduction cost, and total drug and vaccine cost [1]. Over the last three years, feed prices have undergone huge changes. The total feed cost took up approximately 60% of the total cost per pig in China. Thus, feed efficiency is an important component of pig feeding programs. In growing–finishing pig feeding programs, the free-access feeding model is the most common regimen, whereas restricted feeding is generally used in the feeding management of sows [2]. However, previous research has shown/found that pigs with higher feed efficiency visited the feeder fewer times [3]. The lower frequency of daily meal intake improved the conversion ratio of feed into weight gain [4,5]. Time-restricted feeding reduced whole-body fat accumulation and was also associated with inflammation in other research on animals and humans [6,7]. A feeding regimen is an effective way to improve feed conversion efficiency and improve production performance.

The few studies on this issue in growing–finishing pigs have provided some conflicting results in terms of growth performance. Schneider et al. [8] showed that increasing the feeding frequency from two to six times a day significantly improved pig growth performance. Pigs fed a single daily meal (2 h/24) were more efficient in food utilization than nibbling animals [9]. Feeding frequency did not affect the average daily weight gain (ADG) of growing pigs [10]. However, other studies have shown that reducing meal frequency improved the conversion of feed into weight gain [4,5,11]. Moreover, no study has directly evaluated the effect of meal frequency on carcass composition and lipid metabolism in growing–finishing pigs by slaughtering and segmenting to indicate the effect of feeding frequency on lipid metabolism. The aim of this study was to evaluate the effect of meal frequency on growth performance, nutrient digestibility, carcass quality, and lipid metabolism in growing–finishing pigs.

## 2. Materials and Methods

### 2.1. Experiment Design, Animals, and Environment

Sixty-four Duroc × Landrace × Yorkshire barrows and gilts (26.40 ± 2.10 kg initial body weight (BW)) were used in a 112 d experiment in a randomized, completely blocked design. According to initial BW and sex, pigs were randomly allotted to two treatments, eight replicates per treatment, and four pigs (two barrows and two gilts) per pen. The two treatments were free-access to feed group (FA) and three meals per day group (M3), respectively. Daily feed allowance (4 × 5.0% of the BW of the largest pig) of M3 group was calculated at the beginning of the experiment and divided into 3 equal meals that were fed at 06:00, 14:00, and 22:00 each day. If the feed of any pen was fully consumed within one hour, more feed was added to ensure an adequate feed supply. After one hour of feeding, the remaining feed was removed for recording. Dietary nutrient requirements are based on national research council (2012) [12] recommendations, and diets of different phases (Table 1) were provided in a powder form (600 μm). Pigs had free access to water at all times. Pigs were housed in plastic leaky floor pens with one feeder and one nipple drinker. The stocking densities for experiment were 1.00 m^2^ per pig. The relative humidity inside the animal house for all of the experiments ranged from 50% to 70%, and the environmental temperature inside the swine confinement building was controlled at 20 °C.

### 2.2. Growth Performance and Sample Collection

The BW of each pig was measured at 08:00 on days 0, 28, 56, 84, and 112, respectively, to calculate the ADG. The average daily feed intake (ADFI) was measured based on the amount of feed consumed every day. Then, the ratio of gain to feed (G: F) was calculated. Three days before fecal collection, pigs were fed a diet supplemented with 0.5% Cr_2_O_3_ to determine nutrient digestibility. On days 28, 56, 84, and 112, fresh fecal samples were collected directly by massaging the rectum of pigs in each pen to determine digestibility of dry matter (DM), crude protein (CP), and gross energy (GE). The feces were dried in a force-draft oven (65 °C) for 2 d and then weighed, and the dried feces were ground through a 1 mm screen and frozen. Diets and feces were analyzed for DM content by oven drying at 105 °C for 2 h [13]. CP concentration of diets and feces were determined by the combustion procedure using a Leco CHNS-932 Analyzer (Leco Corp., St. Joseph, MI, USA) [13]. GE of diets and feces was determined by using an energy measuring instrument [13]. On day 105, blood collection was completed through the jugular vein of each pig at 07:00 for determination of blood components using a fully automatic hematology analyzer. The Cr_2_O_3_ concentration in diets and fecal samples was determined by spectrometric reading of absorption at 450 nm after wet digestion in nitric acid and 70% perchloric acid [13]. 

At the end of experiment, one pig from each pen was selected randomly for slaughter, with a 1:1 sex ratio for each treatment. Pigs were killed 4 h after their last meal at 06:00 by jugular exsanguination after electronarcosis. After hot carcass weight, dressing percentage, and backfat thickness determination, the left-hand sides of carcasses of slaughtered pigs were divided into shoulder, belly, and ham. Then, each part of the carcass was dissected into retail products, lean trim, fat trim, skin, and bones. Weights of the primal cuts and trim were expressed as proportions of the weight of the left-hand side of each carcass.

### 2.3. Carcass and Meat Quality Analysis

Recording of hot carcass weight was performed after slaughter. Dressing percentage was the ratio of hot carcass weight to slaughter weight. The back fat thickness of all pigs was measured 5 cm from the right-hand side of the midline from 3 different sites to calculate the mean of backfat thickness (shoulder, mid back, and loin at a position directly above the point of the elbow, last rib, and last lumbar vertebra, respectively) Minolta Chromameter CR-300 (Minolta, Osaka, Japan) was used to measure the L*, a*, and b* values of longissimus muscle (LM). The pH probe (pH-STAR; SFK-Technology, Herlev, Denmark) was used to measure the pH (pH_45 min_ and pH_24 h_) of the LM on the right carcass side. Drip loss was determined after LM was suspended at 4 °C for 24 h. Determination of cooking loss was performed after vacuum packaging of LM in a water bath at 70 °C. The Texture Analyzer (TA.XT. plus; Stable Microsystems, Surrey, UK) was used to measure the Warner–Bratzler shear force as described previously [14]. Determination of Intramuscular fat content in the LM was performed using the method described by Folch, 2007 [15].

### 2.4. Lipid Metabolism Analysis

A small amount of backfat samples were collected immediately after slaughter for fat cell diameter determination and determination of lipid-metabolism-related enzyme activity and gene mRNA expression.

Lipid content in intramuscular fat and adipose tissue was determined using methanol–chloroform extraction. The diameter of adipocytes in adipose tissue was determined after hematoxylin and eosin staining.

The enzymatic activity of lipogenesis was determined using the following method. In brief, homogenize after adding 500 mg of LM sample to mixed solution (0.25 mol/L frozen sucrose solution, 1 mmol/L EDTA, and 1 mmol/L dithiothreitol). The mixtures were ultracentrifuged at 100,000× *g* for 1 h at 4 °C; then, the supernatants were collected. Activities of malic enzyme (ME) and glucose-6phosphate dehydrogenase (G6PDH) were measured spectrophotometrically at 340 nm absorbance. Activities of fatty acid synthase (FAS) were measured spectrophotometrically at 340 nm absorbance [16,17].

Real-time fluorescence quantitative (qRT-PCR) amplification was determined of *lipoprotein lipase* (*LPL*), *fatty acid synthase* (*FAS*), *peroxisome-proliferator-activated receptor gamma* (*PPARγ*), *hormone-sensitive lipase* (*HSL*), *perilipin*, *adipocyte differentiation-related protein* (*ADRP*), *leptin and sterol regulatory element binding protein-1* (*SREBP-1*), and other target gene mRNA expressions. Primer sequences are shown in Table 2. qRT-PCR used a 25.0 μL reaction system, including 2.0 μL of cDNA template, 12.5 μL of SYBR green fluorescent dye, 9.5 μL of ultrapure water, 0.5 μL of upstream primers, and 0.5 μL of downstream primers. Using 18S RNA as the internal reference gene, the relative expression of the gene was calculated using the 2^−^^△△Ct^ method [18].

### 2.5. Statistical Analysis

ANOVA analysis of the data was performed using the MIXED program of sas9.4. The feeding frequency was the fixed effect, and sex and BW were the random effects. The pen was used as the experimental unit when analyzing growth performance, nutrient digestibility, and blood profile. The individual pig was used as the experimental unit when analyzing carcass quality, meat quality, and lipid metabolism. Statistical differences were determined at *p* < 0.05.

## 3. Results

### 3.1. Growth Performance, Nutrient Digestibility, and Blood Profile

There was no significant effect of feeding frequency on the final BW, ADG, and nutrient digestibility (Table 3 and Table 4, *p* > 0.05). The ADFI and F: G of the FA group were significantly higher than that of the M3 group during the whole experiment (Table 3, *p* < 0.05). The concentration of leptin and glucose in the blood was not significantly different between treatment groups (Table 5, *p* > 0.05). Reducing meal frequency decreased the concentration of triglycerides, free fatty acids, and urea nitrogen but increased the concentration of insulin in the blood (Table 5, *p* < 0.05).

### 3.2. Carcass Composition and Meat Quality

The slaughter weight did not differ between FA and M3 treatments (Table 6, *p* > 0.05). Hot carcass weight, dressing percentage, and mean backfat thickness were significantly reduced in the M3 group compared with the FA group (Table 6, *p* < 0.05). Reducing the meal frequency decreased the compositions of backfat, belly, and fatty pieces but increased the compositions of ham, LM, and lean pieces in the carcass (Table 6, *p* < 0.05). Feeding frequency had no effect on meat quality traits (Table 7, *p* > 0.05).

### 3.3. Lipid Metabolism

Figure 1 shows that the feeding frequency had no effect on adipose tissue lipid content and fat cell diameter (*p* > 0.05). Greater enzyme activities of ME and FAS and higher mRNA expression of *FAS* and *PPARγ* were found in the LM of the FA group compared with the M3 group (Figure 2 and Figure 3, *p* < 0.05). There was no significant effect of feeding frequency on enzyme activity of G-6-PDH (Figure 2, *p* > 0.05) and mRNA expression of *LPL*, *HSL*, *perilipin*, *ADRP*, *leptin*, and *SREBP-1* (Figure 3, *p* > 0.05).

## 4. Discussion

The results of the present study indicated that reducing meal frequency (M3) of growing–finishing pigs decreased G: F and dressing yield, while no significant changes were found in meat quality traits and nutrient digestibility. Furthermore, blood profile analysis revealed that the components related to protein anabolism were increased, and those related to protein catabolism were decreased. Lipid metabolism analysis showed that the enzyme activities and gene expressions related to adipose anabolism were decreased in M3 pigs compared with FA pigs. In slaughter segmentation, we found that the M3 pigs’ carcasses had a greater lean meat composition.

Growth performance is a key index in growing–finishing pig feeding programs. Satisfaction with various nutritional needs is essential for excellent growth performance [19]. Many studies have been conducted on whether feeding frequency affects feed conversion efficiency. Some experiments showed the opposite results regarding the effects of feeding frequency on pig growth performance [4,5,8,9,10,11,20,21]. Studies indicated that feeding frequency did not affect the growth performance of pigs [22]. Feeding frequency from two to six times daily does not impact the ADFI and ADG of group-housed gilts, and there was no difference in skin and vulva integrity between the gilts and sows [23]. However, other studies showed that reducing feeding frequency (M2 vs. FA) had no significant effect on the ADFI and ADG but could improve the feed-to-meat ratio [10]. Le Naou et al. [4] indicated that pigs fed twice daily improved BW gain and feed efficiency compared with feeding 12 meals per day. Moreover, a study by Schneider et al. [8] showed a significant improvement in pig growth performance after increasing the feeding frequency from two to six times a day. Hua showed the pigs fed one meal per day had a greater ADG than the pigs fed six meals per day [24]. The differences in ADFI between the treatment groups of the above studies contributed to the differences in growth performance [4,9,22]. In the present study, M3 pigs had lower ADFI and G: F compared with FA pigs. No significant difference was observed in the ADG of M3 and FA groups. The reduction in maintenance expenditure due to fewer activities in the M3 group may be the reason for the absence of differences in the ADG compared with the FA group. Although we increased the feeding frequency to three times per day to eliminate the effect of feed intake on growth performance, a lower ADFI was found in M3 pigs. This also illustrated the limited negative effect of feed intake on growth performance regulated by feeding frequency.

Improved nutrient digestibility was thought to be the reason for improved feed conversion efficiency with feeding frequency in some studies [25]. Increasing feeding frequency has been shown to increase the secretion of digestive enzymes in the small intestine and the frequency of secretions in the pancreas [25,26,27,28]. In addition, protein production, chymosin, and lipase activities also increased with increasing feeding frequency [29]. However, we did not observe any differences in nutrient digestibility in the present study. Differences in results of nutrient digestibility may be due to the different studies that used pigs at different stages. Some studies have shown that feeding frequency could affect body weight gain by regulating energy distribution and storage forms [9,30]. Then, we analyzed the blood components and found that insulin and free fatty acid concentrations were significantly increased, but triglyceride and urea nitrogen concentrations were significantly decreased in M3 pigs compared with FA pigs. The release of insulin and amino acids after a meal plays a pivotal role in the stimulation of protein synthesis in skeletal muscle [31,32]. Insulin concentrations in humans need to exceed 10 mU/mL to promote adipogenesis and protein synthesis [33]. The phosphorylation of insulin receptors and stimulation of protein synthesis in skeletal muscle require increased circulating insulin concentrations in neonatal pigs [34]. Therefore, a high concentration of insulin (153.4 pmol/L) in the present study was able to promote more protein synthesis in M3 pigs. Research has also shown that insulin exerts influence on dopamine circuits and reduces the animal’s desire to consume feed [35]. Some studies reported that pigs fed twice daily spent more time resting than those fed ad libitum [36,37]. These results indicated that lower feeding frequency created less energy to walk and was conducive to the improvement in growth performance. In addition, according to the decrease in plasma urea, triglycerides, and free fatty acids concentrations in M3 pigs, it was speculated that reduced meal frequency in growing pigs could reduce the catabolism of protein and amino acids. It has also been found that a lower frequency of eating led to lower concentrations of total cholesterol and triglycerides in other animals and humans [38,39].

In this study, the hot carcass weight and dressing percentage of M3 pigs were lower than that of FA pigs. The increase in gastrointestinal tract weight was due to the larger amount of feed consumed per session for the low-feeding-frequency group [40]. This may be the main reason for the lower slaughter percentage of the M3 group. Computed tomography (CT) scanning was utilized in one study to assess changes in body composition [37]. They found that there was a reduction in the percentage of total carcass fat and an increase in the total percentage of muscle, as determined by CT analysis for the bi-phasic-fed pigs, when compared with those fed ad libitum. In our study, the same results were obtained by slaughter segmentation. However, previous studies showed that body composition was not different between rats or pigs fed 2 vs. 12 meals per day [3,41]. Pigs (30 kg) with a low fat-deposition capacity used in a study showed no effect in terms of body composition (M2 vs. M12), but our study was conducted on fattening pigs measured after 112 days of feeding. Furthermore, we found that the M3 group had significantly reduced enzyme activities in terms of ME and FAS and the mRNA expression of *FAS* and *PPARγ* associated with fat anabolic metabolism. *PPARγ*, a nuclear receptor, is highly expressed in adipose and is now recognized to be a master regulator of adipogenesis [42]. FAS is a key multifunctional enzyme contributing to the synthesis of fatty acids [43]. This further explained the reason why the M3 group had greater lean meat components. Although we found an effect of feeding frequency on lipid metabolism, we did not observe a difference in meat quality. 

In summary, feeding three times a day is a good management method for pig farmers who only sell live pigs, but it is still necessary to carry out cost accounting according to the existing infrastructure conditions and management methods of the farm. For aquaculture enterprises integrating breeding and slaughtering, the negative effects of hot carcass weight that reduce feeding frequency should be fully considered.

## 5. Conclusions

Reducing feeding frequency can improve the conversion of feed into weight gain. Reducing the frequency of feeding changed the pathway of lipid metabolism, which caused a reduction in body fat deposition and increase in muscle mass.

## Figures and Tables

**Figure 1 animals-12-02557-f001:**
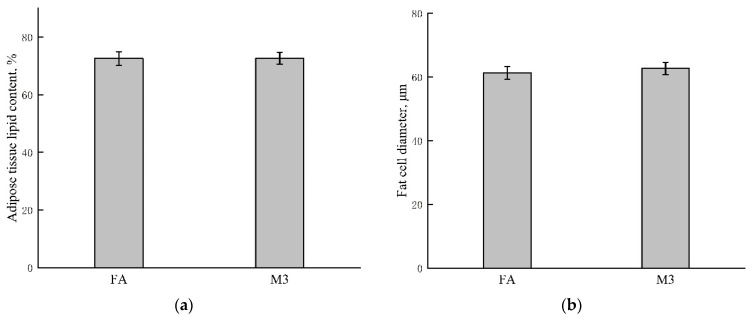
(**a**) Effects of feeding frequency on adipose tissue lipid content; (**b**) Effects of feeding frequency on fat cell diameter. FA, pigs had free access to feed; M3, pigs were given three meals per day (06:00, 14:00, and 22:00 h).

**Figure 2 animals-12-02557-f002:**
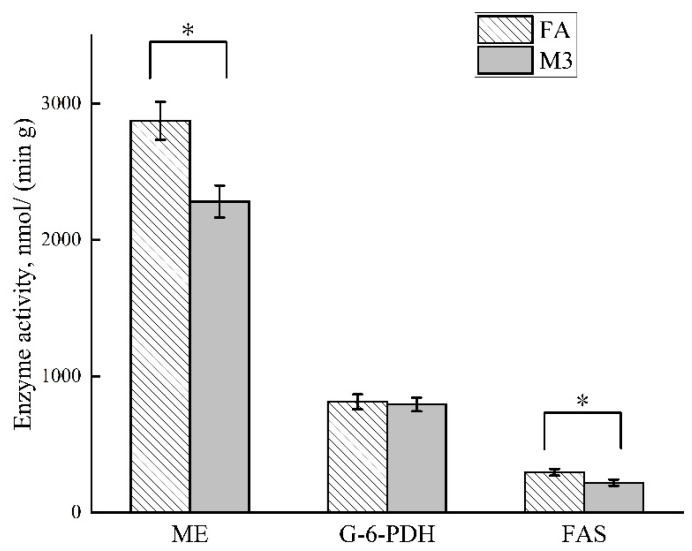
Effects of feeding frequency on enzyme activity of lipid metabolism in growing–finishing pigs. FA, pigs had free access to feed; M3, pigs were given three meals per day (06:00, 14:00, and 22:00 h); * *p* < 0.05; ME, malic enzyme; G-6-PDH, glucose-6phosphate dehydrogenase; FAS, fatty acid synthase.

**Figure 3 animals-12-02557-f003:**
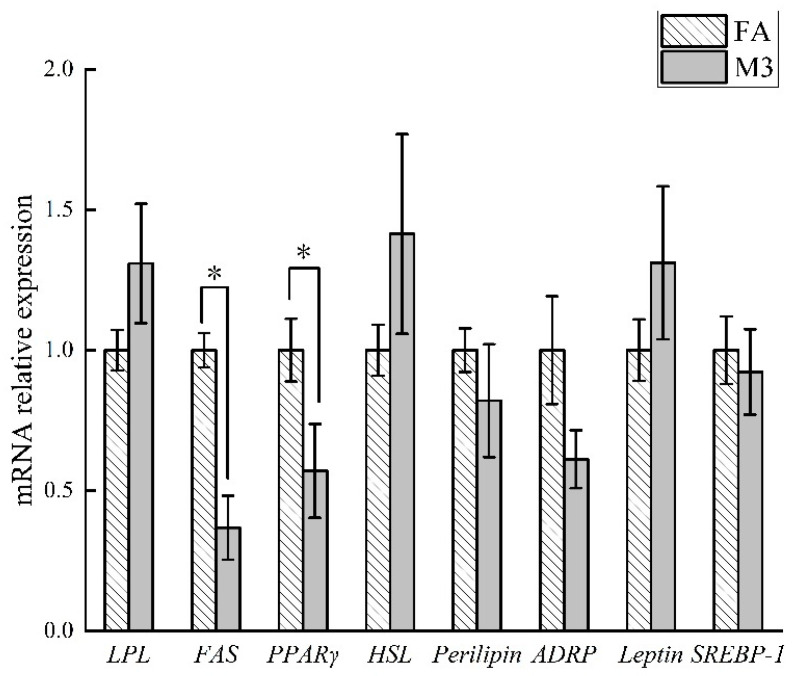
Effects of feeding frequency on mRNA relative expression of lipid metabolism in growing–finishing pigs. FA, pigs had free access to feed; M3, pigs were given three meals per day (06:00, 14:00, and 22:00 h); * *p* < 0.05; *LPL*, *lipoprotein lipase*; *FAS*, *fatty acid synthase*; *PPARγ*, *peroxisome-proliferator-activated receptor gamma*; *HSL*, *hormone-sensitive lipase*; *ADRP*, *adipocyte-differentiation-related protein*; *SREBP-1*, *sterol regulatory element binding protein-1*.

**Table 1 animals-12-02557-t001:** Diets’ composition and nutritional level.

Ingredients, g/kg	Week 1–8	Week 9–12	Week 13–16
Corn	685.0	730.0	761.0
Soybean meal (43% CP)	230.0	180.0	150.0
Wheat bran	40.0	40.0	40.0
Soy oil	15.0	20.0	20.0
Monocalcium phosphate	12.3	11.0	10.0
Limestone (38% Ca)	7..0	7.0	7.0
Salt	3.0	3.0	3.0
Choline chloride (50%)	1.0	1.0	1.0
L-Lys, 78.8%	2.5	3.5	3.5
DL-Met, 98%	0.4	0.5	0.5
L-Thr, 97.5%	1.3	1.5	1.5
Vitamin premix ^1^	1.5	1.5	1.5
Mineral premix ^2^	1.0	1.0	1.0
Total	1000	1000	1000
Determined nutrients			
DE, kcal/kg	3480	3498	3495
Crude protein, %	15.68	13.86	12.79
Ca, %	0.67	0.61	0.58
Total P, %	0.57	0.52	0.49
Ca: P	1.18	1.18	1.18

^1^ Vitamin premix contained per gram of premix: vitamin A, 2640 IU; vitamin D_3_, 264 IU; vitamin E, 17.6 IU; vitamin K activity, 2.4 mg; menadione, 880 μg; vitamin B_12_, 15.4 μg; riboflavin, 3.52 mg; D-pantothenic acid, 8.8 mg; niacin, 13.2 mg. ^2^ Mineral premixes contained per gram of premix: Cu (as copper chloride), 9 mg; I (as ethylenediamine dihydroiodide (EDDI)), 0.36 mg; Fe (as ferrous carbonate), 194 mg; Mn (as manganese oxide), 17 mg; and Zn (as zinc oxide), 149 mg.

**Table 2 animals-12-02557-t002:** Primer sequences.

Genes	Upstream Primer (5′ → 3′)	Downstream Primers (5′ → 3′)	Log in Number	bp
*LPL*	CACATTCACCAGAGGGTC	TCATGGGAGCACTTCACG	NM_214286	177
*FAS*	CTACGAGGCCATTGTGGACG	AGCCTATCATGCTGTAGCCC	NM_001099930	148
*PPARγ*	CCAGCATTTCCACTCCACACTA	GACACAGGCTCCACTTTGATG	NM_214379.1	124
*HSL*	CACAAGGGCTGCTTCTACGG	AAGCGGCCACTGGTGAAGAG	NM_214315	143
*Perilipin*	GCCTGACTTTGCTGGATGG	CTTGGTGCTGGTGTAGGTCTTCT	AY973170	119
*ADRP*	ACATGGCATCCGTTGCTGTT	GGCGTAAGTGTTGGCAATGG	AY621062	251
*Leptin*	CCCTCATCAAGACGATTGTCA	GGTTCTCCAGGTCATTCGATA	AF102856	213
*SREBP-1*	CGCAAGACGGCGGATTTA	GCGACGGTGCCTCTGGTAGT	NM_214157.1	110
*18S RNA*	GATGCGGCGGCGTTATTCC	CTCCTGGTGGTGCCCTTCC	AB117609	125

Note. *LPL*, *lipoprotein lipase*; *FAS*, *fatty acid synthase*; *PPARγ*, *peroxisome-proliferator-activated receptor gamma*; *HSL*, *hormone-sensitive lipase*; *ADRP*, *adipocyte-differentiation-related protein*; *SREBP-1*, *sterol regulatory element binding protein-1*.

**Table 3 animals-12-02557-t003:** Effects of feeding frequency on growth performance in growing–finishing pigs.

Items	FA	M3	SEM	*p*-Values
Average initial body weight, kg	26.36	26.23	0.373	0.645
Average final body weight, kg	119.4	120.1	1.569	0.775
Average daily feed intake, kg				
Day 1–28	1.682	1.616	0.015	0.008
Day 29–56	2.501	2.301	0.060	0.032
Day 57–84	2.849	2.711	0.020	<0.001
Day 85–112	3.334	3.128	0.060	0.026
Day 1–112	2.592	2.439	0.020	<0.001
Average daily weight gain, kg				
Day 1–28	0.766	0.783	0.124	0.361
Day 29–56	0.872	0.884	0.019	0.662
Day 57–84	0.847	0.839	0.010	0.575
Day 85–112	0.834	0.845	0.017	0.649
Day 1–112	0.829	0.838	0.011	0.617
Feed-to-gain ratio				
Day 1–28	2.199	2.067	0.042	0.042
Day 29–56	2.881	2.605	0.083	0.035
Day 57–84	3.365	3.232	0.036	0.024
Day 85–112	4.015	3.715	0.107	0.067
Day 1–112	3.130	2.912	0.046	0.005

Note: FA, pigs had free access to feed; M3, pigs were given three meals per day (06:00, 14:00, and 22:00 h); SEM, Standard error of means.

**Table 4 animals-12-02557-t004:** Effects of feeding frequency on nutrient digestibility in growing–finishing pigs.

Items	FA	M3	SEM	*p*-Values
Dry matter				
Day 28	84.74	85.22	0.521	0.525
Day 56	84.02	83.32	0.466	0.305
Day 84	83.72	83.94	0.471	0.705
Day 112	83.62	83.81	0.389	0.740
Crude protein				
Day 28	83.52	83.61	0.520	0.911
Day 56	79.80	80.63	0.368	0.136
Day 84	78.10	79.10	0.384	0.089
Day 112	77.70	78.57	0.419	0.169
Gross energy				
Day 28	84.50	83.82	0.471	0.323
Day 56	81.74	81.46	0.476	0.678
Day 84	80.86	80.36	0.362	0.197
Day 112	80.69	79.94	0.353	0.153

Note: FA, pigs had free access to feed; M3, pigs were given three meals per day (06:00, 14:00, and 22:00 h); SEM, Standard error of means.

**Table 5 animals-12-02557-t005:** Effects of feeding frequency on blood profile in growing–finishing pigs.

Items	FA	M3	SEM	*p*-Values
Insulin, pmol/L	73.08	153.4	3.806	<0.001
Leptin, μg/L	2.206	2.164	0.025	0.253
Glucose, mmol/L	6.202	5.936	0.104	0.095
Free fatty acids, mmol/L	0.473	0.394	0.017	0.006
Triglycerides, mmol/L	0.549	0.350	0.021	<0.001
Urea nitrogen, mmol/L	4.496	3.469	0.064	<0.001

Note: FA, pigs had free access to feed; M3, pigs were given three meals per day (06:00, 14:00, and 22:00 h); SEM, Standard error of means.

**Table 6 animals-12-02557-t006:** Effects of feeding frequency on carcass quality and carcass composition in growing–finishing pigs.

Items	FA	M3	SEM	*p*-Values
Slaughter weight, kg	117.1	118.0	0.356	0.109
Hot carcass weight, kg	88.28	85.14	0.351	<0.001
Dressing percentage, %	75.37	72.15	0.123	<0.001
Mean backfat thickness, mm	24.99	22.10	0.290	<0.001
Carcass composition, %				
Backfat	8.230	7.630	0.071	<0.001
Belly	13.60	12.80	0.201	0.007
Ham	22.04	23.11	0.277	0.016
Shoulder	24.88	25.45	0.255	0.134
Loin	27.15	27.95	0.288	0.069
Longissimus muscle	5.110	5.200	0.283	0.032
Fatty pieces	21.84	20.35	0.156	<0.001
Lean pieces	74.06	76.51	0.363	<0.001

Note: FA, pigs had free access to feed; M3, pigs were given three meals per day (06:00, 14:00, and 22:00 h); SEM, Standard error of means.

**Table 7 animals-12-02557-t007:** Effects of feeding frequency on meat quality in growing–finishing pigs.

Items	FA	M3	SEM	*p*-Values
pH_45 min_	6.369	6.411	0.039	0.457
pH_24 h_	5.460	5.563	0.048	0.161
Intramuscular fat, %	4.210	4.151	0.056	0.432
Cooking loss, %	35.78	35.33	0.386	0.418
Drip loss, %	4.579	4.428	0.098	0.295
Shear force, kg	4.595	4.660	0.044	0.316
Lightness, L*	47.82	48.98	0.475	0.106
Redness, a*	8.960	9.110	0.070	0.153
Yellowness, b*	3.551	3.474	0.068	0.436

Note: FA, pigs had free access to feed; M3, pigs were given three meals per day (06:00, 14:00, and 22:00 h); SEM, Standard error of means.

## Data Availability

Data will be made available upon reasonable request.

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
