# Peer review of "Reduced Meal Frequency Decreases Fat Deposition and Improves Feed Efficiency of Growing–Finishing Pigs"

_animals, 2022, doi:10.3390/ani12192557_

Round 1

Reviewer 1 Report

This manuscript examined how meal frequency could influence growth performance, nutrient digestibility, carcass quality, and lipid metabolism in growing-finishing pigs. A very well-written manuscript.

General comment:

More application-related discussion is needed in this manuscript. For example, would you suggest pig farmers reduce the meal frequency based on the results? How to balance the positive and negative effects (e.g., decreased hot carcass weight) of reducing the meal frequency?

Specific comments:  

Line 2-3: The title “Reduced Meal Frequency Improved Feed Efficiency of Growing-finishing Pigs”, “feed efficiency” is not able to cover all the studied traits in this research; may consider reversing the title.

Line 36: “….feed prices have suffered huge changes…”. “suffered” is not an appropriate word here; please consider revising the sentence.

Line 38-40: “In growing-finishing pig feeding programs, free-access feeding model is the most common regimen, whereas restricted feeding is generally used in the feeding manage-ment of sows.” References are needed here.

Line 40 - 41: “ However, previous research has established that more feed efficient pigs visited the feeder fewer times”. Consider changing to  “However, previous research has shown/found that pigs with higher feed efficiency visited the feeder fewer times. ”. The words “established” and “more” here make the sentence confusing.

Line 67: “Daily feed allowance (5.0 % of BW) of M3 group was calculated at the beginning of the experiment and divided into 3 equal meals that were fed at 0600, 1400, and 68 2200 h…….”. The same level of Daily feed allowance was used throughout the whole 112-d experiment? Or was it adjusted during the experiment based on the gained BW? The BW used to calculate Daily feed allowance was the individual BW or average BW of pigs fed in the same pen?

Line 95: “At end of the experiment, one pig from each pen was selected randomly for slaughter.” A “the” is needed between “At” and “end”. In the meantime, what was the sex ratio of the sampled individual? Would you consider the barrows and gilts to perform the same in the carcass traits?

Line 153-154: “The pen of 153 pigs served as the experimental unit for statistical analysis of growth performance……”. Do you mean the average values of growth performance traits from the four pigs in each pen were used for statistical analysis? Why not compare the individuals’ values? Or compare the differences among barrows from different pens and the differences of gilts from different pens separately? Because the sex of the pig may need to be considered as a factor during analysis.

Line 168: Table 3. “Initial body weight, kg” & “Final body weight, kg”, you may change to “Average Initial body weight, kg” & “Average Final body weight, kg” if there were the average BW of all pigs at the beginning and the end of the experiment.

Line 223: “Secondly, the growth performance can be improved by other methods.”. This sentence looks like it does not match the following content in the paragraph. Consider removing it.

Line 226-227: “Few studies indicated that feeding frequency did not affect the growth performance of pigs [21]”. Only one reference was cited here, but “few studies” was used in the sentence. May consider revising the sentence.

Line 227-229: “ Feeding frequency from 2 to 6 times daily does not have a negative or positive impact on performance or welfare of group housed gilts and sows [22]”. This sentence is very close to the sentence in the cited paper “Increasing the feeding frequency from 2 to 6 times daily does not appear to have a negative or positive impact on performance or welfare of group-housed gilts and sows”. Need to be revised. Meanwhile, “performance” here need to be more specific (e.g., body condition, aggressiveness, and reproductive failure).

Line 235-236: “The differences in ADFI between the treatment groups of studies contributed to the differences in the final results [3,8,21].” The final results of what? More info is needed here.

Line 240-241: “We were surprised to find no difference in ADG between two treatment groups (M3 vs FA).”. This sentence is unnecessary, and “surprised” here may show readers that the authors had some “bias” during the experiment.  In the meantime, potential reasons why there was no difference in ADG between the two treatment groups are needed to be added.

Line 269-270: “It has also been found that lower frequency of eating leaded to lower concentrations of total cholesterol and triglycerides in other animal and human [37,38].” The references cited here may not be the appropriate ones to support the statement because fasting and timed-restricted feeding, which were used in the cited studies, are different from “lower frequency of eating”. May consider citing better references.

Line 281-282: “Pigs (30kg) with low fat deposition capacity were used in that study, but our study was conducted on…..”. What is the “that study” here? Need to be defined.  

Reviewer 2 Report

The paper was well-written and discussed, but authors need to do the corrections indicated in order the final decision regarding it can be made.

Introduction

Lines 55-56: Please remove the sentence. The phrase is out of place. It can be relocated in Material and Methods section, complementing the already provided information in line 98 if authors consider appropriate since the main research objectives were provided. 

Material and Methods

Line 71: Please relocate the reference [11] after the term: '... NRC (2012) [11]...'

Line 71: Please provide the mean particle size of experimental diets in parenthesis after the term 'powder form'.

Line 92: Re-check this information. Were total Cr-readings: Cr(III) and Cr(VI) determined by AAS (module FAAS) or by ICP-OES / ICP-MS? Correct the term 'spectrophotometric reading' to 'spectrometric reading' in case of AAS or ICP-OES / ICP-MS-analysis of total Cr or correct the term Cr to Cr2O3 in the case of UV-Vis analysis.

Table 1

What %CP of soybean meal was used? Please, add this information in table like you did, for example, to limestone.

Determined nutrients:

Please, specify in Table, if P is the total P or the available P.

Line 139: Please provide a reference (like you did in line 132) that explains in detail the cycling conditions used to perform the PCR analysis.

Results

Line 191: Correct the letter 'l' to the capital letter 'L'.

Discussion

Line 232: Please, correct 'Besides, A study by...' to 'Besides, a study by...'

Line 250: Authors really think they did not find significant differences on nutrient digestibility due to the trial duration? Please explain the statement '...due to the use of pigs at different stages...' What did you mean? Is was not clear.

Line 269: Correct '...frequency of eating leaded...' to 'frequency of eating led...'

Conclusions

Line 293: I suggest authors change '...the way of lipid metabolism...' to '...the pathway of lipid metabolism...'

Round 2

Reviewer 1 Report

The manuscript has improved considerably since the first draft. Overall, it is well written and well organized.

Only one comment:

Line 291-292: I think the word "aquaculture" does not fit here, as "aquaculture" is about fish instead of swine.